# Traditional East Asian Herbal Medicine Treatment for Alzheimer’s Disease: A Systematic Review and Meta-Analysis

**DOI:** 10.3390/ph15020174

**Published:** 2022-01-31

**Authors:** JiEun Lee, Seungwon Kwon, Chul Jin, Seung-Yeon Cho, Seong-Uk Park, Woo-Sang Jung, Sang-Kwan Moon, Jung-Mi Park, Chang-Nam Ko, Ki-Ho Cho

**Affiliations:** 1Department of Korean Medicine Cardiology and Neurology, Graduate School, Kyung Hee University, Seoul 02447, Korea; leeje526@naver.com (J.L.); yahaly@naver.com (C.J.); 2Department of Cardiology and Neurology, College of Korean Medicine, Kyung Hee University, Seoul 02447, Korea; sy.cho@khu.ac.kr (S.-Y.C.); seonguk.kr@gmail.com (S.-U.P.); wsjung@khu.ac.kr (W.-S.J.); skmoon@khu.ac.kr (S.-K.M.); pajama@khu.ac.kr (J.-M.P.); kcn202@khu.ac.kr (C.-N.K.); kihocho58@gmail.com (K.-H.C.)

**Keywords:** Alzheimer’s disease, herbal medicine, traditional East Asian medicine, systematic review, meta-analysis

## Abstract

Alzheimer’s disease (AD) is a leading progressive neurodegenerative disease worldwide, and its treatment is a challenging clinical problem. This review was conducted to evaluate the efficacy and safety of herbal medicine for AD treatment. The PubMed, CENTRAL, EMBASE, CNKI, OASIS, KTKP, and CiNii databases were searched until June 2020 for randomized controlled trials (RCTs) on herbal medicine for AD, and a meta-analysis of 57 RCTs was conducted. For cognitive function, herbal medicine significantly improved the Mini-Mental State Examination (MMSE) and AD Assessment Scale-Cognitive Subscale (ADAS-cog) scores compared with conventional medicine. The MMSE scores showed no significant difference between the groups treated with herbal medicine and donepezil; however, herbal medicine significantly lowered the ADAS-cog score. Acori Graminei Rhizoma-containing and Cnidii Rhizoma-containing herbal medicine significantly improved the MMSE and ADAS-cog scores compared with conventional medicine. Ginseng Radix-containing herbal medicine showed a positive, but not statistically significant, tendency toward improving the MMSE score compared with conventional medicine. Herbal medicine with conventional medicine significantly improved the MMSE, ADAS-cog, and Montreal Cognitive Assessment (MoCA) scores compared with conventional medicine, and herbal medicine with donepezil also significantly improved these scores compared with donepezil. Acori Graminei Rhizoma or Cnidii Rhizoma-containing herbal medicine with conventional medicine significantly improved the MMSE and ADAS-cog scores compared with conventional medicine. Ginseng Radix-containing herbal medicine + conventional medicine significantly improved the MMSE score, but not the ADAS-cog score, compared with conventional medicine. For behavioral and psychological symptoms of dementia, the Neuropsychiatry Inventory (NPI) score was not significantly different between herbal and conventional medicines. Herbal medicine with conventional medicine significantly improved the NPI and Behavioral Pathology in Alzheimer’s Disease Rating Scale scores compared with conventional medicine. The NPI score showed no significant difference between the groups treated with herbal medicine and placebo. Furthermore, herbal medicine with conventional medicine significantly lowered plasma amyloid beta levels compared with conventional medicine alone. Herbal medicine, whether used alone or as an adjuvant, may have beneficial effects on AD treatment. However, owing to the methodological limitations and high heterogeneity of the included studies, concrete conclusions cannot be made.

## 1. Introduction

Dementia is a clinical syndrome that affects memory, thinking, behavior, and activities of daily life, and although 2–10% of cases start before the age of 65, this disease mainly affects older people [1]. In 2018, 50 million people worldwide were diagnosed with dementia, and this number is estimated to triple by 2050 [2]. Alzheimer’s disease (AD) is the most common cause of dementia and accounts for 50–75% of the disease [1].

‘Typical’ late-onset AD is caused by a complex interplay between genetic and environmental factors [3]. The neurodegenerative disease process is characterized by two hallmark pathologies: amyloid beta plaque deposition and neurofibrillary tangles of hyperphosphorylated tau [4]. The mechanisms by which these changes lead to cognitive impairment are still unclear, and several mechanisms have been proposed to explain the pathology of AD. The ‘amyloid cascade’ hypothesis states that amyloid beta aggregation triggers a cascade of events that lead to AD and is currently the dominant pathophysiologic model in AD [3]. However, recent studies show the involvement of complex interactive processes, and research is moving away from the original amyloid hypothesis [5].

Despite the fact that dementia poses significant public health issues, until recently, only five medical treatments that act to control symptoms rather than alter the course of the disease had been approved for AD, involving only two classes of drugs (cholinesterase inhibitors and memantine) [6]. Donepezil, a representative cholinesterase inhibitor, often causes mild adverse effects such as gastrointestinal symptoms, fatigue, and muscle cramps, and in some cases, conduction abnormalities such as sick sinus syndrome [6]. Although memantine has fewer adverse effects than cholinesterase inhibitors, it rarely causes adverse effects such as hypertension, dizziness, headache, lethargy, and constipation [6]. Clinical trials for new treatments failed to show beneficial effects [7]. In June 2021, a new drug that targets the biology of the disease, aducanumab, was approved by the FDA. However, not only is its clinical efficacy uncertain, but it was tested on only early-stage AD patients, and about one-third of the patients experienced adverse events called amyloid-related imaging abnormalities [8].

As traditional East Asian medicine has long been used in treating dementia in East Asian countries, many researchers are now turning to traditional medicines to identify potential neuroprotective or disease-modifying agents [9]. Ukgansan (Yi gan san in Chinese, Yokukansan in Japanese), Palmijihwang-hwan (Ba wei di huang wan in Chinese, Hachimijiogan in Japanese), Gamiondam-tang (Jiawei wen dan tang in Chinese, Kamiuntanto), Dangguijakyak-san (Dangui shaoyao san in Chinese, Tokishakuyakusan in Japanese), and Hwanglyeonhaedok-tang (Huanglain jiedu tang in Chinese, Orengedokuto in Japanese) are a few examples of multi-herb formulas that have been studied for dementia [10]. In addition, a recent study demonstrated that Bojungikgi-tang (Bu zhong yi qi tang in Chinese, Hochuekkito in Japanese) might inhibit amyloid-β aggregation and increase antioxidant activity [11], and multiple human studies have shown that Ukgansan improves the behavioral and psychological symptoms associated with multiple forms of dementia [12]. A randomized, placebo-controlled trial conducted in 2005 reported that Jodeung-san (Diao teng san in Chinese, Chotosan in Japanese) significantly increased the Mini-Mental State Examination (MMSE) and Barthel Index scores in dementia patients [13], and an open-label, crossover trial administering Guibi-tang (Gui pi tang in Chinese, Kihito in Japanese) with acetylcholinesterase inhibitors to AD patients reported that the Japanese version of MMSE (MMSE-J) scores increased significantly during the Guibi-tang intake period [14]. One retrospective analysis showed that compared to conventional therapy alone, adding Chinese herbal medicine had significant benefits in AD treatment, which became more pronounced with time. In the above study, cognitive decline was substantially decelerated in patients with moderate AD, and largely stabilized in patients with mild AD over a period of two years [15]. Furthermore, emerging randomized control trials (RCTs) have continuously reported the effectiveness and safety of herbal medicine for AD. The aim of this study was to analyze various RCTs in order to summarize and evaluate the comparative effectiveness and safety of herbal medicine in clinical trials for AD.

## 2. Methods

### 2.1. Study Registration and Design

This systematic review and meta-analysis evaluated the efficacy and safety of herbal medicine in patients diagnosed with AD. The protocol of this study was registered in the Research Registry (registration number: reviewregistry933) [16].

### 2.2. Database Search

This review was conducted in accordance with the Preferred Reporting Items for Systematic Reviews and Meta-Analyses (PRISMA) guidelines [17] and the Cochrane Handbook for Systematic Reviews of Interventions [18]. The following databases were searched for articles published until June 2020: MEDLINE (PubMed), Cochrane Central Register of Controlled Trials (CENTRAL), Excerpta Medica Database (EMBASE), China National Knowledge Infrastructure (CNKI), Oriental Medicine Advanced Searching Integrated System (OASIS), Korean Traditional Knowledge Portal (KTKP) and Citation Information by National Institute of Informatics (CiNii). In addition, a manual search was performed to search the reference lists of relevant articles and Google Scholar to identify further studies. The search strategies were modified for each database, and there was no language restriction when selecting studies. The detailed search strategy for each database is as in Appendix A.

### 2.3. Eligibility Criteria

The initially screened studies were subsequently reviewed and selected according to the following eligibility criteria: only RCTs were included; non-RCTs, case reports, case series, uncontrolled trials, and laboratory studies were excluded, as were trials that failed to provide detailed results. Patients clinically diagnosed with AD by criteria such as the Diagnostic and Statistical Manual of Mental Disorders, 4th Edition (DSM-IV) or the National Institute of Neurological and Communicative Disorders and Stroke and the Alzheimer’s Disease and Related Disorders Association (NINCDS-ADRDA) were included, regardless of their age, gender, nationality, and educational background. Patients with other disorders that might affect cognitive function, such as stroke, Parkinson’s disease, and traumatic brain injury, were excluded. Patients with any other rare forms of dementia other than AD, such as Lewy Body, frontotemporal, or vascular dementia, were also excluded. Studies using herbal medicine as experimental interventions were included with no limitations on dosage, frequency, and duration of treatment. Various formulations of herbal medicine were allowed (e.g., decoctions, tablets, capsules, powders); however, only studies in which herbal medicine was administered orally were included. Herbal medicine alone and concurrent treatment with conventional therapy were both considered acceptable if herbal medicine was applied only to the treatment group. The control interventions included placebo or conventional medicine.

### 2.4. Outcome Assessment

For the primary outcome, the MMSE score was assessed to evaluate cognitive function. For secondary outcomes, the Alzheimer’s Disease Assessment Scale–Cognitive Subscale (ADAS-cog) and Montreal Cognitive Assessment (MoCA) Test for Dementia scores were assessed for further evaluation of cognitive function. Secondary outcomes also included the Neuropsychiatric Inventory (NPI) and Behavioral Pathology in Alzheimer’s Disease Rating Scale (BEHAVE-AD) scores for behavioral and psychological symptoms of dementia (BPSD) and plasma levels of amyloid-beta. The number and severity of adverse events were also reviewed.

### 2.5. Data Extraction

Two independent reviewers (JEL and SK) extracted following data from the included studies: general information (e.g., authors, title, publication year), study methods (e.g., interventions and comparisons, intervention duration, study design, sample size, randomization details, blinding and any other bias information), participants (characteristics) and outcomes (primary and secondary outcomes, side effects). If there were disagreements between two reviewers, another independent reviewer (KHC) made decisions.

### 2.6. Quality Assessment of Individual Studies

The risk of bias in each included study was assessed using the Cochrane Risk of Bias Tool [19]. Two independent reviewers (JEL and SK) performed the assessment. If there were disagreements between two reviewers, another independent reviewer (KHC) made final decisions. The tool consists of seven sections, including random sequence generation, allocation concealment, blinding of participants and personnel, blinding of outcome assessment, incomplete outcome data, selective reporting, and other biases. Each domain was assessed as “high risk of bias”, “unclear risk of bias”, or “low risk of bias”.

### 2.7. Statistical Analysis

The synthesis was carried out using RevMan V.5.3.5 provided by the Cochrane Collaboration. For continuous data, the pooled results were presented as the mean difference (MD) or standardized MD with 95% confidence intervals (CIs). For dichotomous data, the pooled results were presented as risk ratios with 95% CIs. In terms of heterogeneity, the Higgins I^2^ statistics and Chi-square test were used; significant heterogeneity was thought to exist when the I^2^ value was greater than 50 or the *p* value was less than 0.05.

## 3. Results

### 3.1. Characteristics of Included Studies

A total of 2114 studies were identified by electronic searches. After eliminating duplicates, 1827 studies were left and screened by the abstracts. Subsequently, 115 full-text articles were assessed for eligibility. After reviewing the full texts, 57 studies (4257 patients) published between 2002 and 2020 were included in this review. Fifty-eight studies were excluded for the following reasons: improper interventions (*n* = 10), insufficient data (*n* = 13), improper outcomes (*n* = 8), inclusion of other types of dementia (*n* = 18), no randomization (*n* = 6), duplicate (*n* = 1), not a clinical study (*n* = 1), and crossover trial (*n* = 1) (Figure 1).

Among the 57 studies included, 12 were reported in English [20,21,22,23,24,25,26,27,28,29,30,31] and the remaining 45 studies were reported in Chinese [32,33,34,35,36,37,38,39,40,41,42,43,44,45,46,47,48,49,50,51,52,53,54,55,56,57,58,59,60,61,62,63,64,65,66,67,68,69,70,71,72,73,74,75,76]. The intervention period ranged from 4 to 96 weeks. Eighteen studies did not specifically mention the severity of AD [23,26,28,33,35,37,45,46,47,48,51,58,60,63,67,70,73,74], while the remaining 39 studies mentioned severity varying from mild to severe. The conventional medicine used in control groups also varied and included donepezil [25,27,29,30,31,32,35,36,38,39,40,41,43,44,45,47,48,49,52,54,55,56,57,58,59,60,61,62,63,64,65,67,69,70,71,73,74,75], piracetam [24,26,33,34,37,72], risperidone [20], nimodipine [53], huperzine-A and nicergoline [50], rivastigmine [46], clozapine [51], butylphthalide [68,76], and hydergine [66], among which donepezil was most frequently used.

The type of herbal medicine treatment administered to the treatment groups varied greatly among the studies. Four studies were conducted using a single herb (Korean ginseng [22,23] or *Ginkgo biloba* [25,30]), and the rest used multi-compound preparations. Korean ginseng [22,23], *Ginkgo biloba* [25,30], Yokukansan [20,21], Yizhi Jiannao Granule [58,59,62,75], Bushen Phlegm Stasis Compound [32,33], Bushenyisui Formula [31,43,49], and Bushenyizhi Granule [54,60,67] were each used in two or more studies, and the other studies used different herbal medicines. (“Bushen Phlegm Stastis Compound” is composed of Lycii Fructus, Polygoni Multiflori Radix, Psoraleae Semen, Cistanches Herba, Alpiniae Oxyphyllae Fructus, Astragali Radix, Acori Graminei Rhizoma, Curcumae Radix, Salviae Miltiorrhizae Radix, Cnidii Rhizoma, Pinelliae Rhizoma, Hirudo, and Glycyrrhizae Radix. However, the name of the formula might differ between the studies. “Bushenyisui Formula” is composed of Epimedii Herba, Ligustri Lucidi Fructus, Psoraleae Semen, Radix Polygoni Multiflori, Astragali Radix, Cnidii Rhizoma, and Acori Graminei Rhizoma, and studies might have used different names for this formula.) Among the components of the multi-compound herbal formulas, Acori Graminei Rhizoma [24,26,27,28,31,32,33,34,37,38,39,40,43,45,48,49,51,52,54,55,56,57,60,65,66,67,69,71,74,76], Polygoni Multiflori Radix [24,32,33,34,36,37,42,43,45,46,49,51,57,66,68], Salviae Miltiorrhizae Radix [27,32,33,37,38,40,42,50,51,54,60,61,66,67,68,73,76], Hoelen [20,21,24,28,36,40,41,45,47,52,53,54,55,56,60,67,68,71], Cnidii Rhizoma [20,21,27,29,31,32,33,35,39,42,43,44,45,47,49,50,51,53,54,57,60,66,67,68,76], Epimedii Herba [31,36,37,42,43,45,49,50,54,57,58,59,60,62,63,67,75], Rehmanniae Radix [26,35,36,38,39,40,41,42,45,47,48,51,52,54,55,60,61,66,67,68,71], Ginseng Radix [22,23,24,26,29,37,42,44,45,47,51,61,68,69,72], and Polygalae Radix [24,27,28,39,40,42,51,52,54,55,56,60,65,67,71,74,76] were used in 15 or more studies. The composition details of herbal medicines are listed in Appendix B.

The included studies were divided into three categories. The first category compared herbal medicine with conventional medicine, the second category compared herbal medicine with conventional medicine with conventional medicine, and the third category compared herbal medicine with placebo. Overall, in this review, twenty-five studies compared a herbal medicine group with a conventional medicine group [20,24,26,27,29,31,32,33,34,37,43,45,49,52,53,55,57,58,59,62,64,65,66,71,72,75], among which four studies were performed using the double-dummy technique [29,31,43,57]. Twenty-six studies compared herbal medicine with conventional medicine with conventional medicine [22,23,35,36,38,39,40,41,42,44,46,47,48,50,51,54,56,60,61,63,67,68,69,70,73,76], and two studies compared herbal medicine with placebo [21,28]. Four studies compared three intervention groups. Among them, one study compared two different herbal medicine groups and one conventional medicine group [55], and each herbal medicine was evaluated separately in the “Herbal medicine vs. Conventional medicine” category. Two studies compared herbal medicine, conventional medicine, and herbal medicine with conventional medicine [30,74]. These studies were also evaluated twice, once in the “Herbal medicine vs. Conventional medicine” category and once in the “Herbal medicine + conventional medicine vs. Conventional medicine” category. One study that compared herbal medicine, conventional medicine, and placebo [25] was also evaluated twice; once in the “Herbal medicine vs. Conventional medicine” category and once in the “Herbal medicine vs. Placebo” category. One study of the “Herbal medicine vs. Conventional medicine” category compared herbal medicine with three different conventional medicines (donepezil, rivastigmine, and galantamine) [27], and in this review, herbal medicine was compared with only donepezil, the first-line therapy of AD. One study of the “Herbal medicine + conventional medicine vs. Conventional medicine” category administered two different doses of herbal medicine [22], and the high dose was evaluated in this study.

For cognitive function assessment, MMSE was evaluated in 53 studies [20,22,23,24,25,26,27,28,29,31,32,33,34,35,37,38,39,40,41,42,43,44,45,46,48,49,50,52,53,54,55,56,57,58,59,60,61,62,63,64,65,66,67,68,69,70,71,72,73,74,75,76], ADAS-cog in 25 studies [22,23,27,29,31,32,33,36,38,40,42,45,46,48,49,52,54,56,58,60,69,70,71,73,74], and MoCA in three studies [29,38,63]. For BPSD, NPI was evaluated in seven studies [20,21,28,30,31,35,42] and BEHAVE-AD in six studies [28,36,41,46,47,51]. Plasma amyloid beta was evaluated in three studies [29,46,68]. Details on the study characteristics and herbal medicine are shown in Table 1 and Appendix B.

### 3.2. Risk of Bias Assessment

The Cochrane Risk of Bias tool was used for the risk of bias assessment, and details are shown in Figure 2. For random sequence generation, 32 studies provided information on the randomization method and were considered ‘low risk of bias’ [21,24,25,26,27,28,29,30,31,32,36,37,38,39,40,41,42,45,51,54,55,56,60,61,64,65,67,68,69,71,73,76]. Twenty-four studies did not mention the randomization method and were considered ‘unclear risk of bias’ [20,22,23,33,35,43,44,46,47,48,49,50,52,53,57,58,59,62,63,66,70,72,74,75], and one study randomized the patients according to the consultation order and was considered ‘high risk of bias’ [34]. In terms of allocation concealment, most studies did not specifically mention concealment of allocation prior to assignment and were considered ‘unclear risk of bias’ [20,22,23,24,26,27,32,33,35,36,37,38,39,40,41,42,43,44,45,46,47,48,49,50,51,52,53,54,55,56,57,58,59,60,61,62,63,64,65,66,67,68,69,70,71,72,73,74,75,76]. For blinding of participants and personnel, except for a few double-blind studies [21,28,29,30,31,43,57], most studies were considered ‘high risk of bias’. For blinding of outcome assessment, three studies were non-blind and open-label and considered ‘high risk of bias’ [22,23,27]. Nine studies described outcome assessment blinding and were considered ‘low risk of bias’ [21,25,28,29,30,31,43,57,64], and the remaining studies did not describe outcome assessment blinding and were considered ‘unclear risk of bias’. In terms of incomplete outcome data, seven studies that did not mention the reasons for drop-outs or stated the reasons for drop-out as ‘lack of efficacy’ were considered ‘high risk of bias’ [23,25,27,30,49,52,69]. Otherwise, most studies were ‘low risk of bias’. For selective reporting, three studies in which protocols were available were considered ‘low risk of bias’ [21,23,31]. Another bias was ‘unclear’ for all studies (Figure 2).

### 3.3. Synthesized Results

#### 3.3.1. Cognitive Function

##### MMSE

Herbal Medicine vs. Conventional Medicine

Twenty-eight studies evaluated cognitive function using the MMSE score [20,24,25,26,27,29,31,32,33,34,37,43,45,49,52,53,55,57,58,59,62,64,65,66,71,72,74,75], among which one study compared three intervention groups (two types of herbal medicine and one conventional medicine) and was included twice using the same study name [55], resulting in a total of 29 studies. The pooled results of the 29 studies showed that herbal medicine has a significant effect on improving the MMSE score compared with conventional medicine [MD 0.58, 95% CI (0.12, 1.04), *p* = 0.01] with high statistical heterogeneity (I^2^ = 71%) (Figure 3).

A subgroup analysis was conducted based on herbal medicine formulas used in two or more studies. Among them, Yizhi Jiannao Granule was administered in four studies [58,59,62,75], and it did not have significant beneficial effects compared with conventional medicine [MD 0.51, 95% CI (−0.93, 1.96), *p* = 0.49, I^2^ = 79%]. Bushen Phlegm Stasis Compound was administered in two studies [32,33], where it did not show significant results compared with conventional medicine [MD 0.31, 95% CI (−2.29, 2.92), *p* = 0.81, I^2^ = 55%]. Bushenyisui Formula was administered in three studies [31,43,49], and it significantly improved the MMSE score compared with conventional medicine [MD 1.13, 95% CI (0.90, 1.36), *p* < 0.00001, I^2^ = 0%]. When the above-mentioned three herbal medicines were excluded, herbal medicine did not show significantly improved MMSE scores compared with conventional medicine [MD 0.55, 95% CI (−0.13, 1.23), *p* = 0.11, I^2^ = 72%] (Figure 3, Figure A2a).

An additional analysis was conducted based on studies that used donepezil as the control group. Twenty studies were included [25,27,29,31,32,43,45,49,52,55,57,58,59,62,64,65,71,74,75], and herbal medicine did not significantly improve the MMSE score compared with donepezil [MD 0.30, 95% CI (−0.27, 0.88), *p* = 0.30, I^2^ = 73%] (Figure A1a and Figure A2b).

Studies that included Acori Graminei Rhizoma in the herbal medicine intervention were also analyzed [24,26,27,31,32,33,34,37,43,45,49,52,55,57,65,66,71,74]. Acori Graminei Rhizoma-containing herbal medicine significantly improved the MMSE score compared with conventional medicine [MD 0.94, 95% CI (0.40, 1.49), *p* = 0.0007, I^2^ = 65%] (Figure A1b, Figure A2c). Furthermore, 12 studies that included Cnidii Rhizoma in the herbal medicine intervention were also analyzed [20,27,29,31,32,33,43,45,49,53,57,66]. For MMSE, Cnidii Rhizoma-containing herbal medicine was significantly more effective than conventional medicine [MD 0.73, 95% CI (0.05, 1.41), *p* = 0.03, I^2^ = 69%] (Figure A1c, Figure A2d). Among studies that included Ginseng Radix in the herbal medicine intervention, six studies assessed the MMSE score [24,26,29,37,45,72], and although Ginseng Radix-containing herbal medicine showed beneficial effects compared with conventional medicine, the results were not statistically significant [MD 1.31, 95% CI (−0.03, 2.65), *p* = 0.06, I^2^ = 83%] (Figure A1d).

Another subgroup analysis was conducted based on the treatment period. Among the studies that evaluated MMSE scores between herbal and conventional medicines, four studies had a treatment period of less than 12 weeks [20,53,66,75]. These studies did not show significant differences between the two groups [MD 0.20, 95% CI (−1.44, 1.85), *p* = 0.81, I^2^ = 83%]. The remaining 25 studies had a treatment period of 12 or more weeks [24,25,26,27,29,31,32,33,34,37,43,45,49,52,55,57,58,59,62,64,65,71,72,74], and these studies showed that herbal medicine significantly improves the MMSE score compared with conventional medicine [MD 0.66, 95% CI (0.19, 1.14), *p* = 0.006, I^2^ = 66%] (Figure A1e).

Herbal Medicine + Conventional Medicine vs. Conventional Medicine

Twenty-four studies in this category assessed the MMSE score [22,23,35,38,39,40,41,42,44,46,48,50,54,56,60,61,63,67,68,69,70,73,74,76], and herbal medicine with conventional medicine had a significant beneficial effect compared with conventional medicine [MD 2.79, 95% CI (2.09, 3.48), *p* < 0.00001] with high statistical heterogeneity (I^2^ = 85%). Therefore, a subgroup analysis was conducted. Bushenyizhi Granule was administered in three studies [54,60,67], which showed that Bushenyizhi Granule with conventional medicine significantly improves the MMSE score compared with conventional medicine [MD 3.58, 95% CI (1.68, 5.48), *p* = 0.0002, I^2^ = 44%]. Results were significant even when Bushenyizhi Granule was excluded [MD 2.71, 95% CI (1.97, 3.45), *p* < 0.00001, I^2^ = 86%] (Figure 4, Figure A2e).

An additional analysis was conducted using 17 studies that used donepezil as the control [35,38,39,40,41,44,48,54,56,60,61,63,67,69,70,73,74]. Herbal medicine + donepezil significantly improved the MMSE score compared with donepezil only [MD 2.50, 95% CI (1.75, 3.26), *p* < 0.00001, I^2^ = 82%] (Figure A1f, Figure A2f).

Eleven studies in this category included Acori Graminei Rhizoma in the herbal medicine intervention [38,39,40,48,54,56,60,67,69,74,76], and Acori Graminei Rhizoma-containing herbal medicine with conventional medicine was significantly effective in improving the MMSE score compared with conventional medicine [MD 2.47, 95% CI (1.63, 3.31), *p* < 0.00001, I^2^ = 71%] (Figure A1g, Figure A2g). Additionally, ten studies that included Cnidii Rhizoma in the herbal medicine intervention and evaluated the MMSE score were analyzed [35,39,42,44,50,54,60,67,68,76]. Cnidii Rhizoma-containing herbal medicine with conventional medicine significantly improved the MMSE score compared with conventional medicine [MD 2.74, 95% CI (1.70, 3.79), *p* < 0.00001, I^2^ = 72%] (Figure A1h, Figure A2h). Seven studies that administered Ginseng Radix-containing herbal medicine were analyzed, and Ginseng Radix containing herbal medicine with conventional medicine significantly improved the MMSE score compared with conventional medicine [22,23,42,44,61,68,69] [MD 2.92, 95% CI (1.96, 3.88), *p* < 0.00001, I^2^ = 67%] (Figure A1i).

Another subgroup analysis was conducted based on the treatment period. Among the studies that evaluated the MMSE score between herbal medicine with conventional medicine and conventional medicine, four studies had a treatment period of fewer than 12 weeks [40,41,48,70], and pooled results showed that herbal medicine with conventional medicine significantly improves the MMSE score compared with conventional medicine when administered for a period of fewer than 12 weeks [MD 2.86, 95% CI (1.35, 4.38), *p* = 0.0002, I^2^ = 84%]. In studies with a treatment period of 12 or more weeks, herbal medicine with conventional medicine also significantly improved the MMSE score compared with conventional medicine [MD 2.76, 95% CI (1.94, 3.57), *p* < 0.00001, I^2^ = 85%] (Figure A1j).

Herbal Medicine vs. Placebo

Three studies assessed the MMSE score when comparing herbal medicine with placebo [21,25,28], and there was no significant difference between the two groups [MD 0.22, 95% CI (−0.65, 1.10), *p* = 0.62, I^2^ = 0%] (Figure A1k).

##### ADAS-cog

Herbal Medicine vs. Conventional Medicine

Eleven studies compared the ADAS-cog score between herbal medicine and conventional medicine [27,29,31,32,33,45,49,52,58,71,74]. The pooled results showed that herbal medicine significantly lowers the ADAS-cog score compared with conventional medicine [MD −1.96, 95% CI (−3.61, −0.31), *p* = 0.02, I^2^ = 83%] (Figure 5, Figure A2i). Ten studies in this category used donepezil as the control [27,31,32,33,45,49,52,58,71,74], and herbal medicine significantly improved the ADAS-cog score compared with donepezil [MD −1.87, 95% CI (−3.67, −0.08), *p* = 0.04, I^2^ = 85%] (Figure A1l, Figure A2j).

Nine studies that included Acori Graminei Rhizoma in the herbal medicine intervention were additionally analyzed [27,31,32,33,45,49,52,71,74]. Acori Graminei Rhizoma-containing herbal medicine significantly improved the ADAS-cog score compared with conventional medicine [MD −2.37, 95% CI (−4.15, −0.58), *p* = 0.009, I^2^ = 85%] (Figure A1m). Seven studies that included Cnidii Rhizoma in the herbal medicine intervention were also analyzed [27,29,31,45,49]. Cnidii Rhizoma-containing herbal medicine significantly lowered the ADAS-cog score compared with conventional medicine [MD −3.05, 95% CI (−5.20, −0.89), *p* = 0.006, I^2^ = 83%] (Figure A1n). Two studies in this category included Ginseng Radix in the herbal medicine intervention [28,44], and there was no significant difference between Ginseng Radix-containing herbal medicine and conventional medicine [MD −5.66, 95% CI (−15.42, 4.10), *p* = 0.26, I2 = 94%] (Figure A1o).

Herbal Medicine + Conventional Medicine vs. Conventional Medicine

ADAS-cog was evaluated in 14 studies [22,23,36,38,40,42,46,54,56,60,69,70,73,74], and results showed that herbal medicine with conventional medicine was significantly more effective than conventional medicine [MD −5.34, 95% CI (−7.87, −2.81), *p* < 0.0001] with high statistical heterogeneity (I^2^ = 95%) (Figure 6, Figure A2k).

Among the 14 studies analyzed, 10 used donepezil as the control [36,38,40,54,56,60,69,70,73,74]. Herbal medicine with donepezil significantly improved the ADAS-cog score compared with donepezil only [MD −5.32, 95% CI (−9.50, −1.14), *p* = 0.01, I^2^ = 96%] (Figure A1p), Figure A2l).

Seven studies in this category included Acori Graminei Rhizoma in the herbal medicine intervention [38,40,54,56,60,69,74], and Acori Graminei Rhizoma-containing herbal medicine with conventional medicine significantly lowered the ADAS-cog score compared with conventional medicine [MD −3.10, 95% CI (−4.49, −1.72), *p* < 0.0001] with low statistical heterogeneity (I^2^ = 17%) (Figure A1q). Studies that included Cnidii Rhizoma in the herbal medicine intervention were also analyzed [42,54,60]. For ADAS-cog, Cnidii Rhizoma-containing herbal medicine with conventional medicine was significantly more effective than conventional medicine [MD −8.18, 95% CI (−12.45, −3.92), *p* = 0.0002, I^2^ = 71%] (Figure A1r). Four studies administered Ginseng Radix-containing herbal medicine [22,23,42,69], and there was no significant difference between Ginseng Radix containing herbal medicine with conventional medicine and conventional medicine in terms of the ADAS-cog score [MD −5.01, 95% CI (−10.79, 0.77), *p* = 0.09, I^2^ = 87%] (Figure A1s).

A subgroup analysis was conducted based on the treatment period. Two studies had a treatment period of less than 12 weeks [40,70], and the difference between herbal medicine with conventional medicine and conventional medicine was not statistically significant [MD −12.51, 95% CI (−29.63, 4.61), *p* = 0.15, I^2^ = 98%]. However, herbal medicine with conventional medicine significantly improved the ADAS-cog score compared with conventional medicine when the treatment period lasted for 12 or more weeks [22,23,36,38,42,46,54,56,60,69,73,74] [MD −4.03, 95% CI (−5.35, −2.70), *p* < 0.00001, I^2^ = 77%] (Figure A1t).

##### MoCA

Herbal Medicine + Conventional Medicine vs. Conventional Medicine

MoCA was assessed in two studies [38,63], and herbal medicine with conventional medicine significantly improved the MoCA score compared with conventional medicine [MD 2.29, 95% CI (1.03, 3.55), *p* = 0.0004, I^2^ = 0%]. Both studies used donepezil as control (Figure A1u).

#### 3.3.2. BPSD

##### NPI

Herbal Medicine vs. Conventional Medicine

Three studies compared herbal medicine with conventional medicine to evaluate the NPI score [20,30,31], and there was no significant difference between the two groups [MD −0.49, 95% CI (−1.10, 0.12), *p* = 0.12, I^2^ = 0%] (Figure A1v).

Herbal Medicine + Conventional Medicine vs. Conventional Medicine

Three studies assessed the NPI score [30,35,42], and results indicated that herbal medicine with conventional medicine significantly lowers the NPI score compared with conventional medicine [MD −3.01, 95% (CI −4.35, −1.68), *p* < 0.0001, I^2^ = 0%] (Figure A1w).

##### BEHAVE-AD

Herbal Medicine + Conventional Medicine vs. Conventional Medicine

Five studies in this category assessed the BEHAVE-AD score [36,39,46,47,51], and herbal medicine with conventional medicine showed a significant beneficial effect compared with conventional medicine [MD −2.86, 95% CI (−4.14, −1.59), *p* < 0.0001, I^2^ = 91%] (Figure A1x).

#### 3.3.3. Plasma Amyloid-Beta

Two studies comparing herbal medicine with conventional medicine and conventional medicine evaluated plasma amyloid-beta [46,68], and herbal medicine with conventional medicine significantly lowered plasma amyloid-beta levels compared with conventional medicine alone [MD −9.20, 95% CI (−13.92, −4.47), *p* = 0.0001, I^2^ = 80%] (Figure A1y).

### 3.4. Safety Assessment

Twenty-one studies did not mention adverse events [33,34,36,40,42,43,45,50,53,55,57,58,59,60,62,65,68,70,71,72,75]. Among the other 36 studies, nine studies reported no adverse events in the treatment group [24,25,26,28,39,44,46,52,69]. Twenty-six studies mentioned mild adverse events which included nausea, insomnia, and stomach discomfort [20,22,23,27,29,30,31,32,35,37,38,41,47,48,49,51,54,56,61,63,64,66,67,73,74,76]. Among these studies, five studies stated that the adverse events were significantly lower in the treatment group compared with the control group [20,30,47,61,67]. Three studies mentioned serious adverse events in the treatment group but concluded that these events were not related to the treatment [30,31,49]. One study that administered Yokukansan to the treatment group reported cases of hypokalemia, acute heart failure, and choleslithiasis [21]. However, the control group was placebo, making it difficult to compare the percentage of these events with conventional medicine.

### 3.5. Publication Bias

Funnel plot analyses were conducted when 10 or more studies were included in the analysis to assess publication bias (Figure A2) [18]. All funnel plots were asymmetrically distributed, suggesting potential publication bias.

## 4. Discussion

AD is a growing global health concern and treatments proven to alter the disease pathology are not available [3]. Symptoms usually start with memory loss as the presenting symptom [77] and progress gradually. Behavioral and psychological symptoms are also common and cause significant distress to caregivers and increase mortality in patients [77]. However, there are no licensed treatments for these symptoms [3]. Dementia is a complex and multifaceted disease, and additional therapeutic methods are needed to help patients adapt to the disease’s progress [6].

Amyloid beta accumulation in the brain is thought to initiate the AD process, and during the past 20 years, efforts have been made to develop drugs that affect the formation or clearance of Aβ; however, these treatments have failed to show positive results regarding cognitive function in all past clinical trials [7]. Not only did cognitive outcomes not improve, but for some patients, the symptoms worsened, despite the fact that the treatments reduced brain Aβ levels [78]. Most recently, aducanumab, a human monoclonal antibody that targets Aβ fibrils and oligomers, underwent two phase III trials (EMERGE and ENGAGE). Based on data from these trials, the FDA granted approval to aducanumab on 7 June 2021, the first Alzheimer’s drug to be approved in almost 20 years [79]. However, one review had reported that it could not be concluded that aducanumab has clinical benefits based on the data from these trials [80], and many scientists still say that there is not enough evidence to say aducanumab is an effective treatment for Alzheimer’s [81]. Therefore, alternative potential neuroprotective and disease-modifying agents are needed.

For cognitive function, herbal medicine showed statistically significant effects compared with conventional medicine in the MMSE and ADAS-cog scores with high statistical heterogeneity. Therefore, subgroup analyses were conducted in an attempt to lower heterogeneity, and herbal medicines used in two or more studies were evaluated. While Yizhi Jiannao Granule and Bushen Phlegm Stasis Compound did not have significant effects, Bushenyisui Formula significantly improved the MMSE score compared with conventional medicine with low heterogeneity. Herbal medicine with conventional medicine significantly improved the MMSE, ADAS-cog, and MoCA scores compared with conventional medicine alone. Through a subgroup analysis, Bushenyizhi Granule with conventional medicine showed a significant effect with low statistical heterogeneity compared with conventional medicine in the MMSE score. When herbal medicine was compared with placebo, there was no significant difference between the two groups regarding MMSE. Additional analyses were performed to compare herbal medicine with conventional medicine with the first-line therapy for AD, donepezil. When herbal medicine combined with donepezil was compared with donepezil alone, herbal medicine with donepezil significantly improved the MMSE, ADAS-cog, and MoCA scores. For BPSD, when herbal medicine was compared with conventional medicine, the NPI score was not significantly different between the two groups. Herbal medicine with conventional medicine was significantly more effective than conventional medicine in the NPI and BEHAVE-AD scores. One study compared the NPI score between herbal medicine and placebo, and the score was not significantly different between the two groups.

A large variety of herbs were used in each herbal medicine of the selected studies. Among them, Acori Graminei Rhizoma appeared in 30 studies and Cnidii Rhizoma in 25 studies, and both were included in the two formulas that showed significant results in the subgroup analyses—Bushenyizhi Granule and Bushenyisui Formula. In addition, Ginseng Radix, which appeared in 17 studies, is one of the most popular herbs, with continuously increasing use worldwide [82]. Therefore, additional analyses were conducted to evaluate the efficacy of herbal medicines containing Acori Graminei Rhizoma, Cnidii Rhizoma, or Ginseng Radix. In this review, results showed that herbal medicine that included Acori Graminei Rhizoma or Cnidii Rhizoma significantly improved MMSE and ADAS-cog scores compared with conventional medicine. Similar results were obtained when herbal medicine including Acori Graminei Rhizoma or Cnidii Rhizoma was combined with conventional medicine.

Acori Graminei Rhizoma was first mentioned in Shinnongbonchokyung (神農本草經) as an herb that “treats wind-cold damp impediment, cough and counterflow qi ascent, opens the heart portals, supplements the five viscera, frees the nine orifices, brightens the eyes and sharpens the hearing, and helps the articulation of the voice (主治風寒濕痹, 咳逆上氣, 開心孔, 補五臟, 通九竅, 明耳目, 出音聲)” [83]. Acori Graminei Rhizoma is one of the most frequently cited herbs in terms of memory improvement, forgetfulness, and aging in classical Chinese medical literature [84], and has been used for the treatment of neurodegenerative diseases in Chinese traditional medicine for thousands of years [85]. One of its main components is β-asarone, which is known to have neuroprotective effects via various mechanisms. An experimental study reported that in Aβ42-induced rats, β-asarone ameliorated cognitive impairment and reversed the increase of apoptosis in the hippocampus by inhibiting JNK activation, upregulating Bcl-w and Bcl-2, and inhibiting caspase-3 activity [86]. Another experimental study reported that in Aβ42-induced rats, β-asarone improved cognitive impairment and alleviated Aβ deposition by protecting astrocytes, which was possibly achieved by reducing the levels of TNF-α and IL-1β and then downregulating AQP4 expression [87]. Other authors reported that β-asarone mitigated learning and memory impairments and improved synaptic plasticity by suppressing the excess release of IL-6, IL-1β, i-NOS, and COX-2 in dizocilpine-treated mice [88]. In one study, β-asarone reduced the APP, PS1, Aβ, BACE1, and p62 levels while increasing the SYN1, BECN1, and LC3 levels in PC12 cell AD models, suggesting that β-asarone protects the PC12 cell model against Aβ42 by promoting autophagy [89]. In another study, β-asarone increased the activities of SOD and GPX in Alzheimer rats [90]. Overall, β-asarone has anti-inflammatory, antioxidant, and anti-apoptotic effects and protects neurons from damage by amyloid-beta. Cnidii Rhizoma also first appeared in Shinnongbonchokyung (神農本草經) as an herb that “treats wind entering the brain, headache, cold impediment, hypertonicity of the sinews which are sometimes slack and sometimes tense, incised wounds, and blood block in females (主中風入腦, 頭痛, 寒痺, 筋攣緩急, 金創, 婦人血閉無子)” [83], and is commonly used to treat cardiovascular and neurovascular diseases in traditional Chinese medicine [91]. One of the main components of this herb is tetramethylpyrazine (TMP)—also known as ligustrazine—which has shown therapeutic effects in various neurodegenerative conditions such as ischemic stroke, Parkinson’s disease, and AD [92]. One animal study reported that TMP restores the cAMP levels and, thus, has a protective effect on the cAMP/PKA/CREB signaling pathway, which leads to the reversal of memory deficits induced by scopolamine [93]. Another study reported that by inhibiting caspase-3 activation and increasing the Bcl-2/Bax ratio, TMP protects PC12 cells against H_2_O_2_-induced apoptosis and plays a role in neuroprotection [94]. The complexity and multicausality of dementia have become increasingly recognized in recent studies [95]. The above mechanisms imply that Acori Graminei Rhizoma and Cnidii Rhizoma have multitarget neuroprotective effects and can be potential agents for AD.

Ginseng Radix is one of the most popular, high-value herbs worldwide [82], and is traditionally used to treat weakness and fatigue and to promote longevity [96]. Ginseng Radix was first mentioned in Shinnongbonchokyung (神農本草經) as an herb that “tonifies the five viscera, tranquilizes the essence-spirit, relieves ethereal and corporeal souls, suppresses fright palpitations, eliminates pathogens, improves vision, opens the heart, and enhances intellect. When taken for a long period, it makes the body light and prolongs life (主補五臟，安精神，定魂魄，止驚悸，除邪氣，明目、開心、益智。久服，輕身延年)” [83]. Recently, studies on the use of Ginseng Radix for improving cognitive performance have increased, and Ginseng Radix is known to have antioxidant, anti-inflammatory, and anti-apoptotic effects [82]. Studies have shown that Ginseng Radix has two active components—ginsenosides and ginotonin—which inhibit β- and γ-secretase activity, enhance hippocampal neurotrophic factor expression, attenuate neuroinflammation and mitochondrial oxidative stress, and stimulate non-amyloidogenic α-secretase activity [97]. Furthermore, one experimental study reported that fermented ginseng treatment significantly improves memory function in transgenic mouse models, and significantly reduces the soluble A-β42 levels in the cerebral cortex [98]. In this review, Ginseng Radix-containing herbal medicine with conventional medicine significantly improved the MMSE score compared with conventional medicine. However, Ginseng Radix-containing herbal medicine alone did not show statistically significant effects on MMSE and ADAS-cog scores compared with conventional medicine. These results suggest that Ginseng Radix-containing herbal medicine can be used as adjuvant therapy. Combining Ginseng Radix with an AD drug might ameliorate the symptoms of AD by inducing synergistic effects, enhancing efficacy, and having additional neuroprotective effects [97].

Traditional East Asian medicine emphasizes individuality, with its unique characteristic of “pattern identification,” a traditional approach to classify disease symptoms into patterns [99,100], which helps practitioners select relevant treatments. Based on this theory, different herbal formulas can be used to treat the same disease [100]. It is considered that herbal medicine administered based on pattern identification is more effective; a previous study showed that herbal medicine taken in accordance with the diagnosed pattern is more effective in acute stroke patients [101]. Therefore, in clinical settings, herbal medicine is usually prescribed as multi-herb formulas based on pattern identification. In the included studies of this review, except for two studies that administered Ginseng Radix as a single herb, Acori Graminei Rhizoma, Cnidii Rhizoma, and Ginseng Radix were administered as components of a formula rather than a single herb. It might be suggested that the positive results of this review regarding Acori Graminei Rhizoma, Cnidii Rhizoma, and Ginseng Radix were achieved by synergistic interactions between the pharmacological actions of the herbs and use of pattern identification. Therefore, Acori Graminei Rhizoma, Cnidii Rhizoma, and Ginseng Radix may be beneficial in clinical settings when added to herbal medicine in accordance with pattern identification.

In this study, two studies reported that herbal medicine with conventional medicine intervention significantly lowers plasma amyloid beta levels compared with conventional medicine [46,68]. However, these results must be interpreted with caution. First, these two studies did not mention the evaluated isoforms of amyloid-beta, which is important because Aβ40 and Aβ42 might have different influences on AD pathology [102]. Moreover, previous studies have shown conflicting results regarding plasma amyloid-beta levels in AD patients. One study reported that plasma Aβ42 levels are significantly higher in AD than in healthy subjects [103], while another study reported that AD patients have reduced plasma Aβ42 and Aβ40 levels [104]. Other authors reported that the levels of plasma Aβ42/40 are significantly low in AD patients [105], and one meta-analysis revealed a statistically insignificant variation in plasma Aβ42 levels between AD patients and controls [106]. Therefore, it is difficult to conclude that AD patients have lower Aβ levels than normal subjects. Furthermore, biomarkers might remain dynamic even after the onset of AD. A previous study showed that patients with an advanced stage of dementia (CDR = 1–2) had increased Aβ40 levels compared to those with an early stage of dementia (CDR = 0.5) [107]. In the future, studies are needed not only to compare patients with controls but also to validate the change in plasma amyloid-beta levels based on disease progression [108]. Referring to these previous studies, concrete conclusions cannot be made on whether the results of this review regarding plasma amyloid beta are beneficial. Blood biomarkers are cost-effective and non-invasive compared with positron emission tomography (PET) imaging or cerebrospinal fluid (CSF) biomarkers and can be collected and analyzed routinely to detect or track the disease [109]. Therefore the trend of biomarkers of AD is moving from CSF to blood [110]. In the future, more studies should be conducted to elucidate the changes in the levels of plasma amyloid beta and other plasma biomarkers in the progression of AD.

A total of three studies compared herbal medicine with placebo [21,25,28], and when assessing the MMSE and NPI scores, no significant differences were found between the two groups. Among the three studies, one study stated that for the placebo group, the main cause for withdrawal was “lack of efficacy,” which might have altered the results [25]. Another study that administered Yokukansan to the herbal medicine group was a multi-center trial and reported that the centers that reported strong placebo effects carried out non-pharmacotherapies, and the influence of these therapies cannot be ignored [21]. Furthermore, in this study, Yokukansan significantly improved the “agitation/aggression” and “hallucinations” subcategories of the NPI-Q scale in subgroups with an MMSE score of less than 20 or with an age younger than 74 compared with placebo. Owing to the small number of studies and reasons mentioned above, it is difficult to conclude that herbal medicine is not significantly beneficial compared with placebo. Herbal medicine might still be useful in clinical settings, and further trials with a large sample size and standardization of other therapies are needed.

Three studies compared the NPI score between herbal and conventional medicines [20,30,31], and there was no significant difference between the two groups. However, two of the three studies reported that the herbal medicine group had a significantly lower prevalence of adverse events than the conventional medicine group. Drug safety is an important aspect of long-term disease treatment. A previous pragmatic randomized, open-label trial reported that 81.2% of caregivers reported adverse events of the study medications, and adverse events were the main cause for discontinuing the study drug [111]. Moreover, a meta-analysis reported that all-cause discontinuation and discontinuation due to adverse events were higher with cholinesterase inhibitors than with placebo [112]. Although the differences between herbal and conventional medicines were not statistically significant, herbal medicine might be a safe therapeutic option to consider for BPSD in clinical settings.

This study has some strengths compared with previous meta-analyses. Various interventions and outcomes were investigated. Scales for cognitive function as well as BPSD were investigated, giving a broader perspective on the efficacy of herbal medicine in AD. Databases were searched regardless of the language and publication types and included 57 studies (4257 patients), a number larger than the previous studies, which might increase the validity of the study results. Subgroup analyses were conducted based on herbal medicines used in two or more studies as well as herbal medicines that included Acori Graminei Rhizoma, Cnidii Rhizoma, or Ginseng Radix. Although statistical heterogeneity among the studies could not be resolved, the results of this study might suggest specific herbal medicines that are promising in clinical settings and future research fields. In addition, changes in plasma amyloid-beta levels were analyzed, which has not been done in previous studies.

This systematic review and meta-analysis has several limitations. First, many of the studies showed bias in terms of qualitative research methodology. Many studies did not describe random sequence generation or allocation concealment, and thus selection bias might exist. In addition, the blinding of participants was mostly absent (performance bias), and the blinding of outcome assessors was unclear in most studies (detection bias); these results might have influenced the results of this study. Second, because of the differences in the degrees of dementia, duration of treatment, prescriptions, and doses of herbal medicine, high heterogeneity was observed among the studies. In particular, some studies that were analyzed did not show the severity of AD. This may have possibly affected the evaluation of treatment outcomes and the sake of fair comparison. Third, many studies only reported outcome results measured at the end of treatment, making it difficult to evaluate the long-term effects of herbal medicine in AD patients. Fourth, most of the studies included in this review were conducted in China. Ethnic and sociocultural factors might influence trial outcomes [113]; therefore, it is difficult to extend the results to patients of other ethnic backgrounds. Furthermore, studies conducted in China have a tendency to have high rates of success [114]. Finally, not all studies reported adverse effects and many studies had small sample sizes, making it difficult to draw concrete conclusions.

Although it is difficult to draw concrete conclusions because of methodological limitations of the included studies and high heterogeneity among them, the results of this study indicate that herbal medicine—especially when administered with conventional medicine—may improve the symptoms of AD patients and, therefore, be used as adjuvant therapies in clinical settings to improve the effectiveness and reduce the limitations of conventional medicine. In particular, according to the results of the present study, Bushenyisui Formula, Bushenyizhi Granule (for cognitive decline), and Yokukansan (for BPSD) are noteworthy. However, among the selected studies of the present study, only two studies evaluated plasma amyloid beta. Although the results of the meta-analysis confirmed the possibility that herbal medicine has an amyloid-beta alleviation effect, the amount of research is still insufficient and there is a scientific gap with the current research trend. Therefore, studies on a larger scale and with high quality are needed in future trials, as well as safety evaluations, assessment of amyloid-beta, and analyses of long-term follow-up.

## Figures and Tables

**Figure 1 pharmaceuticals-15-00174-f001:**
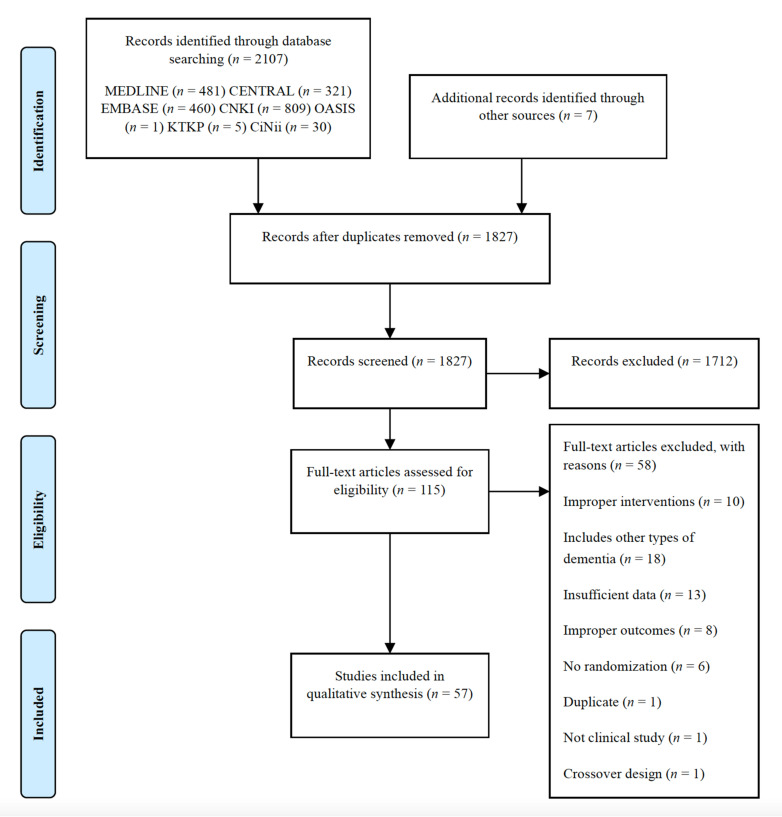
The PRISMA flow chart of study selection.

**Figure 2 pharmaceuticals-15-00174-f002:**
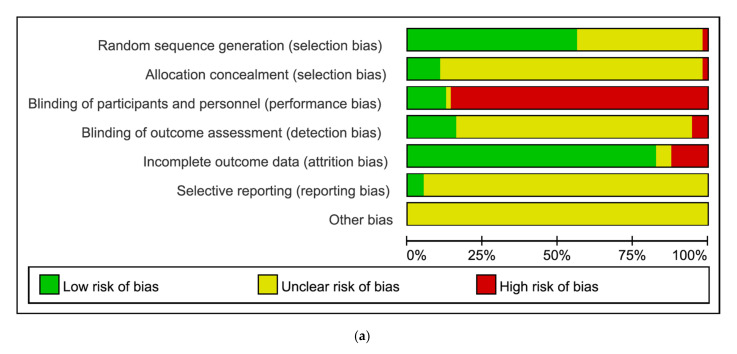
(**a**) Risk of bias graph. (**b**) Risk of bias summary (“+” = low risk of bias, “−” = high risk of bias, “?” = unclear risk of bias.).

**Figure 3 pharmaceuticals-15-00174-f003:**
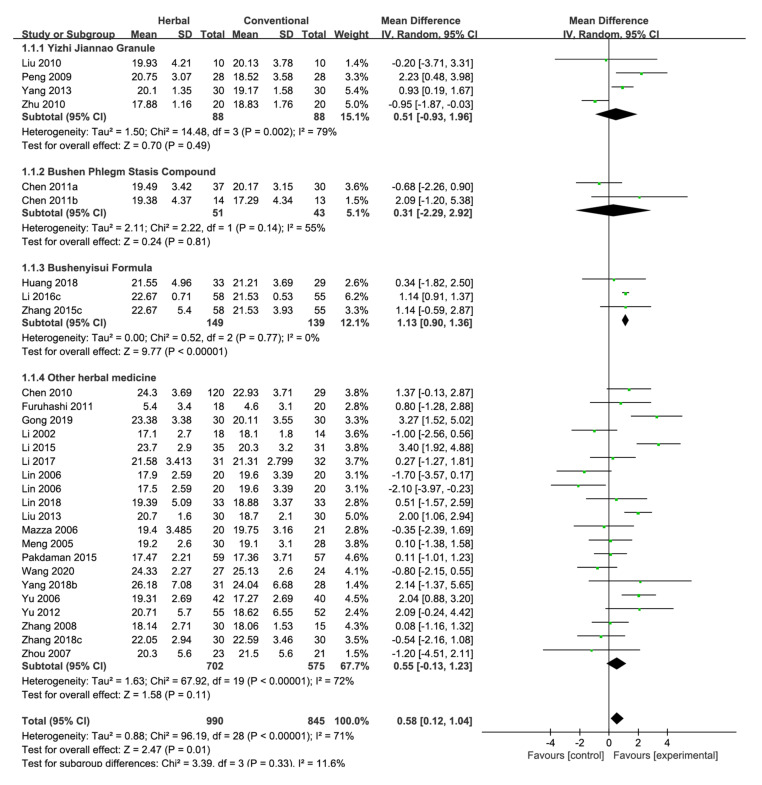
Forest plot for MMSE: Herbal medicine vs. Conventional medicine.

**Figure 4 pharmaceuticals-15-00174-f004:**
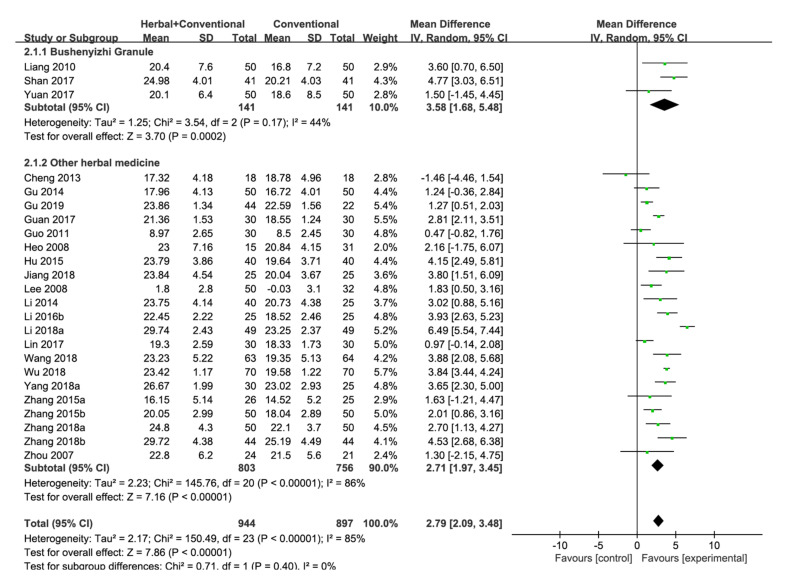
Forest plot for MMSE: Herbal medicine + conventional medicine vs. Conventional medicine.

**Figure 5 pharmaceuticals-15-00174-f005:**
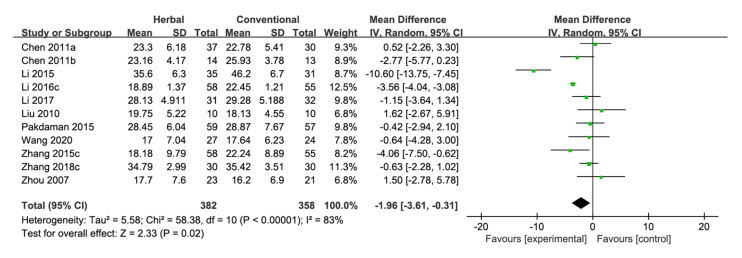
Forest plot for ADAS-cog: Herbal medicine vs. Conventional medicine.

**Figure 6 pharmaceuticals-15-00174-f006:**
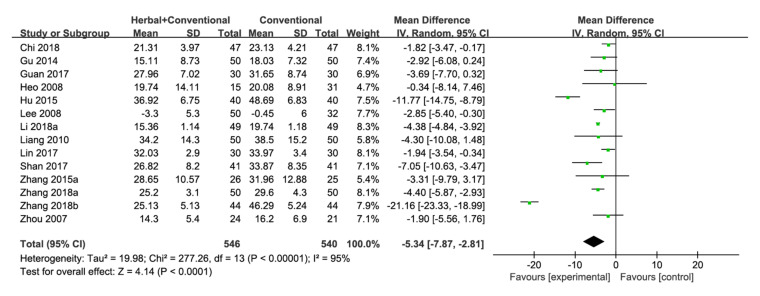
Forest plot for ADAS-cog: Herbal medicine + conventional medicine vs. Conventional medicine.

**Table 1 pharmaceuticals-15-00174-t001:** Summaries of the included studies.

Study	Severity of AD Sample Size (T/C) Mean Age (T/C)	Treatment Period (Weeks)	Intervention of Treatment Group	Intervention of Control Group	Outcomes
Furuhashi 2011 [20]	Severe 38(18/20) 82.8/83.1	4	Yokukansan 22.5 g/d	Risperidone 1 mg/d	Barthel Index, MMSE, NPI *, CMAI *
Furukawa 2017 [21]	Mild to moderate 137(72/65) 78.3/78/5	4	Yokukansan 7.5 g/d	Placebo 7.5 g/d	NPI-Q *, MMSE
Heo 2008 [22]	Mild to moderate 46(15/31) 67.73/66.68	12	Korean red ginseng 9 g/d with usual treatment	Control: usual treatment (Treated with donepezil, rivastigmine, memantine for at least 6 months before randomization)	ADAS (total *, cog *, noncog), MMSE, CDR *
Lee 2008 [23]	N/A 82(50/32) 66.6/65.6	12	Korean white ginseng powder 4.5 g/d + Usual treatment	Usual treatment	MMSE *, ADAS-cog *, ADAS-noncog
Liu 2013 [24]	Mild to moderate 60(30/30) 74/75	12	Bushen-huatan-yizhi Granule 12 g/d	Piracetam 2.4 g/d	MMSE *, ADL *, serum SOD *, LPO *, TG *
Mazza 2006 [25]	Mild to moderate 60 (T:20/G:21/P:19) 66.2/64.5/69.8	96	*Ginkgo biloba* 160 mg/d	1. Donepezil 5 mg/d 2. Placebo	MMSE, SKT *, CGI *
Meng 2005 [26]	N/A 58 (30/28) 70.2/68.4	12	Naohuandan 12 capsules/d	Piracetam 4.8 g/d	MMSE *, ADL *
Pakdaman 2015 [27]	Mild to moderate 116(59/57) 71.8/71.8	64	MLC601 (NeuroAiD^TM^) 3 capsules/d	Donepezil according to clinical response and recommended maximum or tolerable dose.	MMSE, ADAS-cog
Pan 2014 [28]	N/A 91(45/46) 57.2/56.9	20	Shen-Zhi-Ling Oral Liquid 30 cc/d	Placebo 30 cc/d	BEHAVE-AD, NPI, DFA of actigraph activity, MMSE
Wang 2020 [29]	Mild to moderate 51(27/24) 65.48/60.88	24	Jiannao Yizhi Formula 10 g/d with Donepezil placebo 5 mg/d	Donepezil 5 mg/d with Jiannao Yizhi Formula placebo 10 g/d	ADAS-cog *, MMSE *, MoCA *, ADL, CMSS *, Serum Ach *, Aβ42 *, Tau *
Yancheva 2009 [30]	Mild to moderate 88(30/29/29) 69/66/68	22	1. *Ginkgo biloba* 240 mg/d with Donepezil placebo 2. *Ginkgo biloba* 240 mg/d with (Donepezil 5 mg/d for the first month, 10 mg/d for the remaining period)	Donepezil 5 mg/d for the first month, 10 mg/d for the remaining period with *Ginkgo biloba* extract placebo	SKT, CDT, Verbal fluency test, NPI (total, distress), GBS (total, ADL), HAM-D, CSS (Tinnitus, Dizziness)
Zhang 2015c [31]	Mild 113(58/55) 72.79/72.97	24	Yishen Huazhuo Decoction 100 mL/d with Donepezil placebo 5 mg/d	Donepezil 5 mg/d + Yishen Huazhuo Decoction placebo 100 mL/d	ADAS-cog *, MMSE, ADL, NPI
Chen 2011a [32]	mild to moderate 67(37/30) 73.89/75.27	12	Bu Yuan Cong Nao Tang 100 mL/d	Donepezil 5 mg/d	MMSE *, ADAS-cog *, ADL *
Chen 2011b [33]	N/A 27(14/13) 72.36/74.38	12	Bushen Phlegm Stasis Compound 100 mL/d	Piracetam 2.4 g/d	MMSE *, ADAS-cog *, ADL *
Chen 2010 [34]	Mild to moderate 149(120/29) 66.78/68.29	12	Compound Polygonum multiflorum extract 30 mL/d	Piracetam 2.4 g/d	MMSE *, ADL *
Cheng 2013 [35]	N/A 36(18/18) 70.33/68.56	12	Yang xue qing nao Granule 12 g/d with (Donepezil 5 mg/d for the first 2 weeks, 10 mg/d for the remaining period)	Donepezil 5 mg/d for the first 2 weeks, 10 mg/d for the remaining period	MMSE, NPI *
Chi 2018 [36]	Mild to severe 94(47/47) 75.36/74.17	24	Bushen Tianjing Yisui Prescription 1 dose/d with (Donepezil 5 mg/d for the first month, 10 mg/d for the remaining period)	Donepezil 5 mg/d for the first month, 10 mg/d for the remaining period	ADAS-cog *, SIB *, BEHAVE-AD *, ADL *, HAMA *, CSDD *
Gong 2019 [37]	N/A 60(30/30) 73.32/72.46	12	“Reinforcing Kidney and Removing Blood Stasis” Formula 300 mL/d	Piracetam 3.6 g/d	MMSE *, FAQ *
Gu 2014 [38]	Mild to moderate 100(50/50) 67.3/66.9	24	Dihuang Yizhi Formula 1 pouch/d with Donepezil 5–10 mg/d	Donepezil 5–10 mg/d	MMSE *, MoCA *, ADAS-cog *, ADL *
Gu 2019 [39]	Mild to severe 66(44/22) 76.18/78.73	12	Yizhi Chidai Prescription 400 mL/d with Donepezil 5 mg/d	Donepezil 5 mg/d	MMSE *, TCM symptoms *
Guan 2017 [40]	Mild to severe 60(30/30) 69.3/70.2	8	Di Yong Yizhi Granule 1 dose/d with Donepezil 5 mg/d	Donepezil 5 mg/d	MMSE *, ADAS-cog *, ADL *
Guo 2011 [41]	Moderate to severe 60(30/30) 73.8/73/14	4	Zhibai Dihuang Tang 1 dose/d with Donepezil 5 mg/d	Donepezil 5 mg/d	MMSE *, BEHAVE-AD *
Hu 2015 [42]	Mild to moderate 80(40/40) 68.4/69.2	24	Bushen Tongluo Decoction 1 dose/d with (Donepezil 10 mg/d + Piracetam 2.4 g/d)	Donepezil 10 mg/d with Piracetam 2.4 g/d	MMSE *, ADAS-cog *, ADL *, NPI *, serum SOD *, MDA *, TNF-α *, IL-1 *, IL-6 *
Huang 2018 [43]	Mild 62(33/29) 72.3/72.8	24	Bushenyisui Formula 1 dose/d with Donepezil placebo	Donepezil 5 mg/d + Bushenyisui formula placebo	MMSE *
Jiang 2018 [44]	Mild to moderate 50(25/25) 67.2/70.4	12	Jiannao Powder 10 g/d with (Donepezil 5 mg/d for the first month, 10 mg/d for the remaining period)	Donepezil 5 mg/d for the first month, 10 mg/d for the remaining period	MMSE *, ADL *, HDS-R *, CSF (Aβ42, tau, p-tau) *, TCM symptom score *
Li 2015 [45]	N/A 66(35/31) 69.1/68.4	24	Bupi Yishen Decoction 1 dose/d	Donepezil 10 mg/d	MMSE *, ADAS-cog *, ADL *, TCM symptoms *, serum SOD *, MDA *, T-AOC *, 8-iso-PGF2α *, ox-LDL *
Li 2018a [46]	N/A 98(49/49) 74.45/74.27	24	Compound Congrong Yizhi Capsule 3.6 g/d with (Rivastigmine: First month 3 mg/d, second month 6 mg/d. If well tolerated, increased to 12 mg/d)	Rivastigmine: First month 3 mg/d, second month 6 mg/d. If well tolerated, increased to 12 mg/d	MMSE *, ADAS-cog *, BEHAVE-AD *, ADL *, serum Aβ *, IL-6 *, IL-1β *, BK *
Li 2016a [47]	N/A 98(49/49) 67.5/67.6	12	Renshen Bujing Anshen Formula with Donepezil 20 mg/d	Donepezil 20 mg/d	BEHAVE-AD *, ACE-R *, Barthel Index *
Li 2016b [48]	N/A 50(25/25) 65.60/66.29	8	Huang Cong Formula with Donepezil 5 mg/d	Donepezil 5 mg/d	ADAS-cog, TCM symptoms, MMSE *, FAQ *, HDS *, ADL *
Li 2016c [49]	Mild 113(58/55) 72.79/72.97	24	Tonifying Kidney Prescription 1 pouch/d	Donepezil 5 mg/d	MMSE *, ADAS-cog *, ADL *
Li 2014 [50]	Mild to moderate 65(40/25) 73.55/74.12	12	Bushen Huoxue Decoction 1 pouch/d with (Huperzine-A 200 μg/d + Nicergoline 20 mg/d)	Huperzine-A 200 μg/d with Nicergoline 20 mg/d	MMSE *, ADL *
Li 2018b [51]	N/A 100(50/50) 69.87/68.19	12	Bushen Yizhi formula 600 mL/d with (Clozapine started with 25 mg/d and gradually increased to 200 mg/d)	Clozapine started with 25 mg/d and gradually increased to 200 mg/d	BEHAVE-AD *, serum MDA * and SOD *
Li 2017 [52]	Mild to moderate 63(31/32) 74.82/75.06	12	Modified Shuyu Pill 30 mL/d	Donepezil 5 mg/d	MMSE *, ADAS-cog *, ADL *, SDSD *
Li 2002 [53]	Mild to moderate 32(18/14) 66/65	8	Danggui Shaoyao San 1 dose/d	Nimodipine 60–120 mg/d	HDS *, MMSE, ADL, CGI
Liang 2010 [54]	Mild to severe 100(50/50) 72.6/71.7	24	Bushenyizhi Granule 1 dose/d with (Donepezil 5 mg/d for the first month, 10 mg/d for the remaining period)	Donepezil 5 mg/d for the first month, 10 mg/d for the remaining period	ADAS-cog *, MMSE *, ADL *, TCM symptoms
Lin 2006 [55]	Mild to moderate 60(20/20/20) 76.37/74.43/72.63	12	1. Taioxin 20 mL/d 2. Bushen 20 mL/d	Donepezil 5 mg/d	1.Taioxin: MMSE *, ADL, FOM *, BD, DS *, RVR * 2.Bushen: MMSE *, ADL, FOM, BD *, DS, RVR *
Lin 2017 [56]	Mild to moderate 60(30/30) 62.33/64.57	24	Phlegm-resolving Orifice-opening Decoction 1 dose/d with Donepezil 5 mg/d	Donepezil 5 mg/d	MMSE *, ADAS-cog *, ADL *, TCM symptoms *
Lin 2018 [57]	Mild 66(33/33) 72.29/72.82	24	Bu Shen Yi Sui Decoction + Donepezil placebo	Donepezil 5 mg/d with Bu Shen Yi Sui Decoction placebo	MMSE, ADL, TCM symptoms *, endocrine (testosterone, cortisol *, estradiol, thyroxine, growth hormone *, ACTH *)
Liu 2010 [58]	N/A 20(10/10) 61.5/62.3	12	Yizhi Jiannao Granule 16.5 g/d	Donepezil 5 mg/d	MMSE *, ADAS-cog *, serum IL-1β *, TNF-α *
Peng 2009 [59]	Mild to moderate 56(28/28) 67.2/67.5	12	Yizhi Jiannao Granule 16.5 g/d	Donepezil 5 mg/d	MMSE *, ADL *
Shan 2017 [60]	N/A 82(41/41) 67.32/68.02	24	Bushen Yizhi Granules 1 pouch/d with (Donepezil 5 mg/d for the first month, 10 mg/d for the remaining period)	Donepezil 5 mg/d for the first month, 10 mg/d for the remaining period	MMSE *, ADAS-cog *, ADL *, TCM symptom *
Wu 2018 [61]	Mild to moderate 140(70/70) 70.0/71.0	24	Compound Huonaoshu Capsule 6 granules/d with Donepezil 10 mg/d	Donepezil 10 mg/d	MMSE *, ADL *, EEG signals *
Yang 2013 [62]	Mild to moderate 60(30/30) 84.22/82.72	12	Yizhi Jiannao Granule 11 g/d	Donepezil 5 mg/d	MMSE *, ADL *, homocysteine *
Yang 2018a [63]	N/A 55(30/25) 64.93/63.44	24	Ma Huang Fu Zi Xi Xin Tang 1 dose/d with Donepezil 10 mg/d	Donepezil 10 mg/d	MMSE *, MoCA *, ADL *
Yu 2012 [64]	Mild to moderate 107(55/52) 74.7/76.4	48	Herbal medicine based on syndrome differentiation	Donepezil 5 mg/d	MMSE *, FOM *, BD *, DS *, fMRI
Yang 2018b [65]	Mild 59(31/28) 72.66/73.12	12	Tiaoxin 1 pouch/d	Donepezil 5 mg/d	MMSE *, FOM *, RVR *, BD *, DS *
Yu 2006 [66]	Mild to moderate 82(42/40) Mean 69	10	“Reinforcing kidney, activating blood and resolving phlegm” Formula 1 pouch/d	Hydergine 3 mg/d for the first week, 6 mg/d for the remaining period	MMSE *, ADL *
Yuan 2017 [67]	N/A 100(50/50) 70.2/70.4	24	Bushen Yizhi Granule 1 dose/day with (Donepezil 5 mg/d for the first month, 10 mg/d for the remaining period)	Donepezil 5 mg/d for the first month, 10 mg/d for the remaining period	MMSE *
Zhang 2015b [68]	Moderate to severe 100(50/50) 70/69	12	Shen Gui Yizhi Formula 1 dose/d with Butylphthalide 0.6 g/d	Butylphthalide 0.6 g/d	MMSE *, ADL *, serum Aβ *, TCM symptoms *
Zhang 2015a [69]	Mild to moderate 51(26/25) 67.50/68.72	12	Yizhi Xingnao Fang 1 dose/d with (Donepezil 5 mg/d for the first month, 10 mg/d for the remaining period)	Donepezil 5 mg/d for the first month, 10 mg/d for the remaining period	ADAS-cog *, MMSE *, ADL *, SDSD *
Zhang 2018b [70]	N/A 88(44/44) 72.91/72.87	4	Qingxin Yizhi Decoction 1 dose/d with Donepezil 10 mg/d	Donepezil 10 mg/d	MMSE *, ADAS-cog *, ADL *, serum SOD *, MDA *, T-AOC *, ox-LDL *
Zhang 2018c [71]	Mild to moderate 60(30/30) 65.2/66.4	12	Dihuang Yinzi Decoction 1 dose/d	Donepezil 5 mg/d	MMSE *, ADAS-cog *, ADL *, Levels of Notch1 *, ADAM10 *, BASE1 *
Zhang 2008 [72]	Moderate to severe 45(30/15) Mean 73.5	12	Shenghuang-yizhi Granule 4 pouches/d	Piracetam 3.6 g/d	MMSE *, BBS *
Zhang 2018a [73]	N/A 100(50/50) 70.8/69.2	12	Compound Danshen Tablet 2.88 g/d with (Donepezil 5 mg/d for the first month, 10 mg/d for the remaining period)	Donepezil 5 mg/d for the first month, 10 mg/d for the remaining period	MMSE *, ADAS-cog *, ADL *, serum Glu * and Asp *
Zhou 2007 [74]	N/A 68(23/21/24) 76.1/74.8/75.4	24	1. Reinhartdt and Sea Cucumber Capsule 2.7 g/d 2. Reinhartdt and Sea Cucumber Capsule 2.7 g/d + Donepezil 5–10 mg/d	Donepezil 5–10 mg/d	MMSE *, ADAS-cog, ADL *, thyroid hormones
Zhu 2010 [75]	Mild to moderate 40(20/20) Mean 72.3	8	Yizhi Jiannao Granule 16.5 g/d	Donepezil 5 mg/d	MMSE *, 8-IPF2α (blood, urine) *
Wang 2018 [76]	Mild to severe 127(63/64) 67.3/68.6	12	Bushen Huatan Yizhi Decoction 1 dose/d + Butylphthalide 800 mg/d	Butylphthalide 800 mg/d	MMSE *, hematocrit *, whole blood high shear rate *, whole blood low shear rate *, erythrocyte aggregation index *, MDA *, SOD *, GSH-Px *, CAT *

-Note: This table only shows the interventions included in this review. The names of the herbal medicines were written according to the original paper. If the medicine was written only in Chinese, it was translated to English based on pronunciation. -*: Outcomes that significantly improved in the treatment group compared with baseline. -Abbreviations: AD: Alzheimer’s disease; MMSE: Mini-Mental State Examination; NPI: Neuropsychiatry Inventory; CMAI: Cohen-Mansfield Agitation Inventory; ADAS-Cog: Alzheimer’s Disease Assessment Scale-Cognitive Subscale; CDR: Clinical Dementia Rating; ADL: Activities of Daily Living; SOD: superoxide dismutase; LPO: lipid peroxide; TG: triglyceride; SKT: Syndrom Kurz test (Short Cognitive Performance Test); CGI: Clinical Global Impression; BEHAVE-AD: Behavioral Pathology in Alzheimer’s Disease Rating Scale; DFA: Detrended Fluctuation Analysis; MoCA: Montreal Cognitive Assessment; CMSS: Chinese Medicine Symptom Scale; CDT: Clock-Drawing Test; GBS: Gottfries–Bråne–Steen Scale; HAM-D: Hamilton Rating Scale for Depression; CSS: Concomitant Symptoms Scale; SIB: Severe Impairment Battery; HAMA: Hamilton Anxiety Scale; CSDD: Cornell Scale for Depression in Dementia; FAQ: Functional Activities Questionnaire; MDA: malondialdehyde; TNF-α: Tumor necrosis factor-α; IL: interleukin; HDS: Hasegawa Dementia Scale; CSF: cerebrospinal fluid; TCM: Traditional Chinese Medicine; AOC: antioxidant capacity; 8-iso-PGF2α: 8-iso-prostaglandin F2α; ox-LDL: oxidized low-density lipoprotein; BK: bradykinin; ACE-R: Addenbrooke’s Cognitive Examination-Revised; SDSD: Syndrome Differentiation Scale for Dementia; FOM: Fuld Object-Memory Evaluation; BD: Block Design; DS: Digit Span; RVR: Rapid Verbal Retrieval; ACTH: adrenocorticotropic hormone; EEG: electroencephalogram; fMRI: functional magnetic resonance imaging; BBS: Blessed Behavior Scale; CAT: catalase; GSH-Px: Glutathione peroxidase.

## Data Availability

No new data were created or analyzed in this study. Data sharing is not applicable to this article.

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
