# Peer review of "Traditional East Asian Herbal Medicine Treatment for Alzheimer’s Disease: A Systematic Review and Meta-Analysis"

_pharmaceuticals, 2022, doi:10.3390/ph15020174_

Round 1

Reviewer 1 Report

Lee et al. reviewed the efficacy and safety of Traditional East-Asian herbal medicine for Alzheimer’s disease (AD) treatment by analyzing various randomized control trials (RCTs). This review paper suggests that herbal medicines may have beneficial effects on AD treatment. The manuscript is scientifically and technically sound. The manuscript may be recommended for publication subject to fix the following minor concerns.

  1. Correct the font size in Page No. 1, Line 33.
  2. In the introduction section, briefly add notes on the adverse effects of current treatments for AD.
  3. Letters in the Figures are blurred. Provide Figures 1 to 6 with high resolution.
  4. Give space between words and brackets. Also check journal format for the style of bracket to cite references in text, [ ] or ( ).
  5. There are some grammatical and typographical errors in this manuscript.
  6. The conclusion section needs to be specific. It is important to highlight the achievements and specific scientific gaps in the current trend.
  7. In addition to the conclusion section, the authors should provide detailed suggestions on the future perspectives of herbal medicine for AD treatment.
  8. Italicize the botanical names ex. Ginkgo biloba (Line 183 and 184), Table 1 – Page No. 8, 9, 27, Reference No. 25, 30,
  9. In this review, can the authors recommend any herbal formula with high efficacy and safety for the treatment of AD?

Author Response

Responses for Reviewer 1 ’s Comments
1) Correct the font size in Page No. 1, Line 33.
Thank you for your kind and proper pointing out. As you pointed out, we revised relevant phrase
2) In the introduction section, briefly add notes on the adverse effects of current treatments for AD.
Thank you for your kind and proper pointing out. As you pointed out, we added the adverse effects of current treatments for AD
3) Letters in the Figures are blurred. Provide Figures 1 to 6 with high resolution.
Thank you for your kind and proper pointing out. We created Figures 1 6 in high resolution and inserted them anew.
4) Give space between words and brackets. Also check journal format for the style of bracket to cite references in text, [ ] or ( ).
Thank you for your kind and proper pointing out. We have revised the format of the entire reference citations
5) There are some grammatical and typographical errors in this manuscript.
Thank you for your kind and proper pointing out. As you pointed out, we reviewed and corrected errors found in the entire manuscript.
6) The conclusion section needs to be specific. It is important to highlight the achievements and specific scientific gaps in the current trend.

Thank you for your kind and proper pointing out. As you pointed out, the content of the conclusion in the last paragraph of the discussion has been reinforced. The limitations and complementary points of existing studies were presented in more detail (for the lack of evaluation of amyloid beta). In addition, the effects of herbal medicine were presented in terms of cognitive decline and BPSD.

7) In addition to the conclusion section, the authors should provide detailed suggestions on the future perspectives of herbal medicine for AD treatment.
Thank you for your kind and proper pointing out. We reinforced the points to be addressed in future studies. It emphasized the need for evaluation of amyloid beta and long term follow up.
8) Italicize the botanical names ex. Ginkgo biloba (Line 183 and 184), Table 1 Page No. 8, 9, 27, Reference No. 25, 30,
Thank you for your kind and proper pointing out. We have revised it according to your comments.
9) In this review, can the authors recommend any herbal formula with high efficacy and safety for the treatment of AD?
Thank you for your kind and proper pointing out. As you pointed out, we have added the names of prescriptions worth noting, broadly divided into cognitive decline and BPSD.

Reviewer 2 Report

  1. Page 3. In the database search section, did authors confirm that the Keyword in English and the Keyword in Chinese differ when they accessed the China National Knowledge Infrastructure (CNKI)?
  2. Page 5. Figure 1. Record excluded n=1712. What are the criteria for exclusion?
  3. Herbal medicines should list the producer or manufacturer. Even the medicines with the same name can differ in composition or concentration. On page 17, for instance, Bushenyizhi Granule. There are some variations in the prescription, such as Bushen-Huatan-Yizhi formula.
  4. Page 36. Appendix 1 Search Strategy. 3. CNKI
    1. The keywords for Alzheimer Disease (AD) are not limited to #1 to #5, The “失智症” will be more common in Chinese and Taiwan.
    2. #7 中草葯 is more suitable for Herbal medicine rather than 中葯.

          To get more precise results, I recommend authors try alternative keywords.

Author Response

Responses for Reviewer 2’s Comments
1) Page 3. In the database search section, did authors confirm that the Keyword in English and the Keyword in Chinese differ when they accessed the China National Knowledge Infrastructure (CNKI)?

Yes. In the case of CNKI, there are many papers written only in Chinese, so we conducted a search in Chinese by confirming that the maximum search results can be obtained when searching in Chinese.
2) Page 5. Figure 1. Record excluded n=1712. What are the criteria for exclusion?
Thank you for your kind and proper pointing out. This is the result of screening through Abstract. You can check the related contents in “3.1. Characteristics of included studies”.
3) Herbal medicines should list the producer or manufacturer. Even the medicines with the same name can differ in composition or concentration. On page 17, for instance, Bushenyizhi Granule. There are some variations in the prescription, such as Bushen-Huatan-Yizhi formula.
Thank you for your kind and proper pointing out. Based on your comments, we re-searched all literature and added manufacturers to appendix 2. In the case of prescribing at a medical institution other than a pharmaceutical company, it was judged that there was no manufacturing company and was not filled out.
4) Page 36. Appendix 1 Search Strategy. 3. CNKI
The keywords for Alzheimer Disease (AD) are not limited to #1 to #5, The “失智症 ” will be more common in Chinese and Taiwan.
#7 中草葯 is more suitable for Herbal medicine rather than 中葯 .
To get more precise results, I recommend authors try alternative keywords.

Thank you for your kind and proper pointing out. However, we did not use the broader term "失智症失智症" because our study only covered Alzheimer's disease. Also, referring to other existing studies, "中草葯中草葯" was not used in the search term. Please understand this.